# Color as an Indicator of Properties in Thermally Modified Scots Pine Sapwood

**DOI:** 10.3390/ma15165776

**Published:** 2022-08-21

**Authors:** Magdalena Piernik, Magdalena Woźniak, Grzegorz Pinkowski, Kinga Szentner, Izabela Ratajczak, Andrzej Krauss

**Affiliations:** 1Department of Woodworking Machines and Fundamentals of Machine Design, Faculty of Forestry and Wood Technology, Poznań University of Life Sciences, Wojska Polskiego 38/42, 60-637 Poznań, Poland; 2Department of Chemistry, Faculty of Forestry and Wood Technology, Poznań University of Life Sciences, Wojska Polskiego 75, 60-625 Poznań, Poland

**Keywords:** thermal modification, *Pinus sylvestris* L., color, mechanical properties, ATR-FTIR, chemical composition

## Abstract

The aim of this study was to determine the dependencies between mechanical properties of modified wood and its color. Within its scope, quantitative changes in color and chemical composition (mass loss, total carbon content, content of extractives and main components of wood), as well as mechanical properties (compressive strength along the grain, strength and modulus of elasticity in longitudinal tension tests, compression across the grain and impact resistance) of the modified Scots pine sapwood, were determined. Modifications were conducted in the atmosphere of superheated steam (time—4 h, temperature of 130, 160, 190, 220 °C). Thermal modification of wood results in an increase in the modulus of elasticity, a reduction of elasticity, longitudinal tensile strength and compressive strength perpendicular to grain. It was found that color parameters ∆E, ∆L and ∆a are linear functions of the modification temperature. The existence of functional dependencies between mass loss, longitudinal tensile strength, radial modulus of elasticity and parameters of ∆E and ∆L makes it possible to determine these properties of modified wood based on color. In turn, chemical analysis indicated that an increase in the temperature of wood modification caused a decrease of holocellulose and hemicelluloses contents, especially in wood samples modified at 220 °C.

## 1. Introduction

The thermal modification of wood leads to many significant changes in its chemical structure affecting its subsequent physical, mechanical and technological properties. Basic structural components of wood are degraded, with cellulose and lignin being degraded at a slower rate and at higher temperatures than hemicelluloses. Extractive substances are also broken down to volatile products apart from hemicelluloses [1,2]. Qualitative and quantitative changes occurring in the chemical composition of thermally modified wood are manifested among other things in mass loss, a darkening wood color [3,4,5], as well as reduced hygroscopicity, equilibrium moisture content and swelling in the thickness of wood [6,7,8].

The color of modified wood is an important qualitative characteristic and depends on treatment intensity (temperature, duration and the course of the modification process), thus it may be an indicator of the degree of thermal modification in wood [7,9,10]. Thermal treatment parameters also determine changes in the mechanical properties of modified wood. An increase in temperature above 150 °C leads to decreased longitudinal tensile strength and bending strength, as well as increased hardness, modulus of elasticity, brittleness and toughness [10,11,12]. If based on a change in wood color, it is possible to determine the intensity of thermal treatment, and the mechanical properties of modified wood also depend on the parameters of this treatment, and it may be assumed that wood color is a good descriptor of mechanical properties in thermally modified wood. 

However, the results of studies conducted to date in this respect do not provide grounds for a definite confirmation of this hypothesis. One of the probable causes seems to be related to the fact that studies are generally conducted on material differing in moisture content.

The higher the modification temperature, the darker the wood becomes. At the same time, its equilibrium moisture content decreases. This is manifested in higher strength values recorded in the tests. If this is not taken into account, a controversial observation can be drawn that wood modified at low temperature (light-colored wood—light brown and with relatively high moisture content) has a lower strength than wood modified at high temperature (very dark wood—dark brown and with low moisture content), and the “strength–color” relationship will be increasing. In other words, wood subjected to intensive thermal treatment, the effect of which is the degradation of all chemical components and strong darkening, would be characterized by higher strength than poorly/slightly modified or control wood? In this sense, the study is part of the discussion whether color may be a predictor of the strength of heat-treated wood and of properties of heat-treated versus control wood.

Johansson and Morén (2006) [7], based on the results of bending strength and impact strength testing of modified birch wood, stated that color is not an adequate indicator of wood strength. However, those authors did not obtain uniform color over the entire cross-section of tested samples, which indicates that it may have not been modified over its entire volume, and this may have caused low correlations of the analyzed dependencies. Jämsä and Viitaniemi (2001) [13] pointed out the possible application of various thermal modification processes for wood, thus resulting in diverse properties of modified wood. For this reason, they were of an opinion that, based on changes in the color of modified wood, it may not be concluded to what extent its other properties change in relation to control wood. 

Brischke et al. (2007) [9] indicated the potential applicability of color measurements in the quality control of modified wood, showing a strong correlation between color parameters and mass loss observed during thermal treatment of pine, spruce and beech wood. Bekhta and Niemz (2003) [12] and González-Peña and Hale (2009) [14] showed that some mechanical properties of modified wood may be predicted based on changes in its color parameters. 

To date, very few studies have been devoted to a description of relationships between the mechanical properties of thermally modified wood and its color. The identification of such relationships is significant for the potential application of color measurements to assess the technological quality of modified wood. The aim of this study was to quantitatively determine changes in color and chemical composition, as well as the mechanical properties of thermally modified pine sapwood (*Pinus sylvestris* L.), according to the ThermoWood^®^ procedure, while also identifying dependencies between selected mechanical properties and the color of the modified wood.

## 2. Materials and Methods

### 2.1. Materials

This study determined the selection of the material and the adopted research methodology. In order to minimize the effect of factors related to the anatomical structure of wood at the level of the macro-, micro- and ultrastructure on the resulting properties of thermally modified wood and control wood, it was decided to conduct experiments on possibly the most homogeneous and defect-free material. For this reason, wood for analyses came from the same zone of the cross-section and the height of the stem. Tested wood properties were determined on twin samples of identical moisture content.

The material was sampled from outer zones from a plank of 63 mm in thickness, cut above the breast height diameter from the butt end of a tree aged approx. 100 years. The average width of annual growth rings in this part of the stem cross-section was 2.2 mm, while the share of late wood in the annual growth increments was 34.1%. The density of Scots pine sapwood used in the tests was 530 kg/m^3^.

From defect-free wood, tangentially oriented slats were obtained, which were next cut longitudinally into two parts, in this way producing pairs of twin battens. Next they were planed to transverse dimensions of 20 mm × 20 mm and cut into sections of 330 mm in length. The sections were conditioned until reaching a moisture content of approx. 7%, then half of them were thermally modified and the others constituted the control. Sections were used to obtain samples with shapes and dimensions adequate for tests of the mechanical properties of wood.

### 2.2. Methods

#### 2.2.1. Wood Thermal Treatment Method

The modification of wood in the atmosphere of superheated steam was conducted following the ThermoWood^®^ procedure [15]. Modified sections were heated until, over their entire mass, a temperature of 110 °C was obtained, and this temperature was maintained for 2 h, until the wood moisture content reached approx. 1%. Next the temperature was increased until the assumed value, and then it was maintained for 4 h. At that moment, the temperature of 130 °C inside the sections’ superheated steam was introduced to the chamber. Upon the completion of the wood heating stage at constant temperature, the heating source was turned off, and, upon a reduction of wood temperature to 130 °C, the influx of steam was stopped, and sections were left in the chamber to equalize wood temperature with the ambient temperature. The temperature inside the sections was controlled using thermocouples placed in the middle of their length, height and width, as well as in the chamber above the batch of these sections. The applied procedure made it possible to obtain a uniform temperature over the entire volume of the samples, particularly during the 4 h of modification. The course of the modification process is shown in Figure 1 on an example of a modification at a temperature of 220 °C [16,17].

#### 2.2.2. Physical Properties

Wood color was determined according to the PN-ISO 7724-3:2003 [18] standard using a DataColor TOOLS 600 spectrophotometer (Datacolor Company, Lawrenceville, NJ, USA) recording coordinates in the CIELab system. Measurements were taken on the radial surface of sections, always in the same four sites before and after modification. This procedure was followed on the twin sections. Results were expressed in the number of units of total color difference—∆E. Additionally, the difference in lightness ∆L and chromatic coordinates ∆a and ∆b were determined.

Wood moisture content was determined according to ISO 13061-1 (2014) [19], while density was recorded according to ISO 13061-2 (2014) [20]. Mass loss (WL) was recorded, accurate to 0.1%, as a difference in the weight of absolutely dry wood before and after modification expressed in the percentage of absolutely dry wood mass before modification.

#### 2.2.3. Mechanical Properties

Mechanical properties were determined following the recommendations of respective standards using a ZWICK Z050TH universal strength-testing machine equipped with a ZWICK 066550.02 extensometer (Zwick/Roell, Ulm, Germany). The machine software apart from the graphic plotting of the test made it also possible to calculate the values of strength and modulus of elasticity (MOE). Values of longitudinal compressive and tensile strength were calculated (by the software) as a quotient of force at failure to the surface area of the sample cross-section, while values of MOE were calculated as a quotient of an increment in stresses to strains.

In order to provide an objective assessment of the effect of thermal modification on changes in the mechanical properties of wood, it was decided to determine them both on modified and control samples with comparable moisture content. The control samples were conditioned in desiccators over saturated solutions of lithium chloride (Sigma Aldrich, Darmstadt, Germany) and potassium acetate (Sigma Aldrich, Darmstadt, Germany) at a temperature of 21 ± 1 °C until the moisture content of modified samples was reached. The moisture content of all samples was also controlled directly after the tests. The mean equilibrium moisture content of modified wood at a temperature of 130, 160, 190 and 220 °C was 5.5, 5.1, 4.5 and 3.5%, respectively, while the mean moisture content of the control samples for 130 and 160 °C was 5.0%, while for 190 and 220 °C, it was 3.9%.

Longitudinal compressive strength (CS_L_) was determined according to ISO 13061-17:2017 [21], while compressive strength perpendicular to the grain (so-called stress at proportionality limit), in the tangential direction CS_T_ and the radial direction CS_R_ was tested according to ISO 13061-5:2020 [22].

Longitudinal tensile strength (TS_L_) was determined according to PN-D-04107:1981 [23], in double-sided shoulder samples of 16(T) mm × 20(R) mm × 160(L) mm, narrowed in the central part at a length of 40 mm to a tangential dimension of 2.5 mm. During compression tests, the modulus of elasticity (MOE) was also determined. 

The impact strength (IS) of wood was determined according to PN-D-0404:1979 [24] in rectangular samples of 10(T) mm × 10(R) mm × 110(L) mm using a Louis Schopper pendulum impact machine (support spacing of 70 mm) (Werkstoffprufmaschine Leipzig GmbH, Leipzig, Germany).

#### 2.2.4. Chemical Analysis

The wood samples were mechanically disintegrated to sawdust (particle size below 0.50 mm) and used to determine the contents of the wood’s main components. The samples were extracted in ethanol according to the TAPPI method (T 204 cm-07) [25] in order to obtain the content of extractives. Cellulose content was determined according to the Seifert method [26] using a mixture of acetylacetone, 1,4-dioxane and concentrated hydrochloric acid (Sigma Aldrich, Darmstadt, Germany). Lignin content was analyzed according to the TAPPI method (T 222 om-06) [27] using concentrated 72% sulfuric acid (Sigma Aldrich, Darmstadt, Germany) to hydrolyze and dissolve polysaccharides, while holocellulose content was assessed using sodium chlorite (TAPPIT 9 wd-75) [28]. Hemicelluloses content was calculated based on the difference between the contents of holocellulose and cellulose. 

Carbon content in wood samples was measured using a Flash 2000 elemental analyzer (Thermo Fisher Scientific, Waltham, MA, USA) according to EN ISO 16948:2015 [29]. 

Changes in wood structure after thermal modification were assessed using attenuated total reflectance Fourier transform infrared spectroscopy (ATR-FTIR). The spectra of wood were recorded using a Nicolet iS5 spectrophotometer (Thermo Fisher Scientific, Waltham, MA, USA) recorded over the range of 4000–400 cm^−1^, at a resolution of 4 cm^−1^ and 16 scans.

## 3. Results and Discussion

### 3.1. Wood Color

The color of thermally modified wood darkens with an increase in temperature. This is shown in Figure 2. Variation of color at the level of 2–3 units is considered to be the threshold of perception for color change [30]. The most marked change is observed for samples exposed to the temperature of 220 °C. In the case of lower modification temperatures, wood discoloration is less intensive.

Within the temperature range of 130–220 °C, an increase may be observed in ∆E by 64%, a three-fold reduction in ∆L and an increase in ∆a by 61%. An increase in temperature from 130 to 190 °C does not cause a significant change in the value of ∆b, while elevation of temperature to 220 °C reduces the value of this parameter by 25%, as shown in Table 1. Table 1 presents mean values and the standard deviation (±σ_n−1_) of the recorded parameters.

Values of parameters ∆E, ∆L and ∆a change in proportion to an increase in the modification temperature; these dependencies were approximated using linear functions. The value of ∆b in the temperature range of 130–190 °C remains more or less constant, while an increase in temperature to 220 °C lowers its value. This is shown in Figure 3.

It seems that the change in total color difference (∆E), in wood subjected to thermal treatment within the temperature range of 130–220 °C, first of all results from the coordinate of the difference in lightness ∆L, the value of which changes twice. This is manifested as a darkening of wood color, as well as coordinate ∆a, the value of which changes by approx. 60%, causing a stronger exposure of the red color in wood. This observation is confirmed by testing results [9,14,31].

Strong dependencies between total color difference (∆E) and the difference in the lightness (∆L) of wood and the temperature of its modification make it possible not only to predict but also to obtain the required color of wood subjected to thermal treatment, as well as to determine the temperature used in the modification process based on color parameters (Figure 3).

Relative values of color parameters change as follows: parameter ∆E from 1.2 (130 °C) to 2.0 (220 °C) and parameter ∆L from 1.2 to 3.7, parameter ∆a from 1.3 to 2.1, respectively. In turn, parameter ∆b is on average 1.18 (130 to 190 °C), while, at the temperature of 220 °C, it is 0.85. Relative value = value of the tested property of modified wood at a given temperature referred to the value of this property recorded for twin samples of the control wood.

### 3.2. Mechanical Properties

Regardless of the adopted modification temperature (130–220 °C), longitudinal compressive strength (CS_L_) of modified wood is 68–76 MPa, with a mean of 73 MPa, and it is comparable to the strength of control wood, in which the mean value is 75 MPa. It is shown in Table 2, which presents mean values and the standard deviation (±σ_n−1_) of the recorded parameters.

The lack of a relationship between wood modification temperature and longitudinal compressive strength is also confirmed by the analysis of the absolute values of these parameters. It is presented in Figure 4.

Thermal treatment does not cause a reduction of the longitudinal compressive strength of wood. This is probably cause by the specific character of wood failure in the longitudinal compression test. At the submicroscopic level, wood may be treated as a composite reinforced with fibers. Wood strength along the grain is the strength of the cell walls (lumens do not transfer loads) and is determined by the collapse of axial elements of its structure (a specific case of buckling)—similarly to composites, which fail during axial compression [32,33,34]. 

At the final stage of wood failure during longitudinal compression, cell walls are deformed and delaminated [35]. The obtained results suggest that the process of wood failure at compression along the grain is initiated at such low stresses that the negative effect of thermal modification, observed in the case of other mechanical properties of wood, is not manifested in this case. The initiation of wood failure at a low level of compressive stresses and the character of the failure process [34,36,37] may be the reason for the lack of any manifested effect of thermal treatment on the longitudinal compressive strength of wood. Moreover, in thermally modified wood, no loss was observed in the content of lignin, which stiffens cell walls and is their component most resistant to the action of temperature [38,39].

Wood modification in the temperature range of 130–220 °C causes no changes in the level of stresses at the limit of proportionality at tangential compression (CS_T_); their mean value is 5.3 MPa, while for control wood, it is 7.1 MPa (Table 2). Thermal treatment run at a temperature of 130–220 °C causes a steady, approx. 25% reduction in CS_T_ in relation to control wood (Figure 4). 

An increase in temperature above 160 °C causes a decrease in the values of stress at the limit of proportionality in the radial compression test (CS_R_) from 3.5 to 2.7 MPa (Table 2). The modification temperature of 190 °C results in a decrease by 13%, while the temperature of 220 °C by 21% in relation to control wood (Figure 4).

In wood during the compression test perpendicular to the grain, thermal treatment results in a reduction of CS_T_ by, on average, 25% and CS_R_ by 9%. A greater decrease in the strength in the tangential direction compared to the radial direction indicates that thermal modification to a greater extent affects changes in the properties of late wood rather than early wood, since mainly late wood is responsible for the modification of the mechanical properties of wood in the tangential direction in relation to annual rings.

An increase in modification temperature within the range from 130 to 220 °C results in a considerable reduction of the modulus of elasticity in the tangential and radial directions, amounting to 27 and 46%, respectively. The mean values of MOE in modified wood amount to 700 MPa for the tangential direction and 280 MPa for the radial direction, while for control wood, it is 440 and 220 MPa, respectively (Table 2). Compared to control wood, treatment at 130 °C results in a 1.8-fold increase in MOE_T_ and MOE_R_. With an increase in modification temperature, their values decrease, which is particularly evident in the radial direction (Figure 4).

The mean value of MOE_L_ for wood modified at temperatures of 130–190 °C was 18.9 GPa, while, at a temperature of 220 °C, it is 16.2 GPa; whereas in control wood, it is 17.9 GPa (Table 2). Modification at a temperature of 130–190 °C results in an increase of MOE_L_ on average by 7% in relation to control wood, which shows the increased rigidity of the material, which indicates changes in the chemical structure of wood components, primarily cellulose. These observations confirm the reports that an increase in thermal modification temperature leads to an increase in the modulus of elasticity, hardness and brittleness, as well as susceptibility to cracking [10,40,41]. In turn, a reduction of the relative value of MOE_L_ by 12% probably results from a drastic lowering of hemicelluloses content in wood subjected to thermal treatment at 220 °C.

No significant change was observed in the longitudinal tensile strength (TS_L_) of wood modified at a temperature of 130 and 160 °C compared to control wood, while modification at 190 and 220 °C caused a considerable reduction of TS_L_ by 40 and 50%, respectively (Figure 4). This reduction of TS_L_ values is explained by changes in the chemical structure shown in our experiments, i.e., a considerable decrease in the content of hemicelluloses and an increased share of lignin in modified wood, as confirmed by research [10]. The mechanical properties of modified wood are positively correlated with the content of hemicelluloses (mainly pentosanes and hexosanes) and negatively correlated with lignin [10].

The impact resistance of modified wood decreased from 3.8 to 1.5 J/cm^2^, while that of control wood is 4.0 J/cm^2^ (Table 2). With an increase in modification temperature, relative values of impact resistance decrease, at the temperature of 220 °C reaching only approx. 40% impact resistance of control wood. Within the temperature range of 190–220 °C, a drastic reduction of impact resistance is observed (Figure 4). 

A decrease in longitudinal tensile strength, as well as strength at dynamic loading in the case of modified wood, indicates an increase in its brittleness, which is particularly evident for treatment at temperatures of 190–220 °C. These observations confirm reports of a considerable reduction of the impact resistance of thermally modified wood [10,11] and indirectly indicate changes in the wood structure at temperatures above 150 °C, which is significant in terms of mechanical properties [39].

### 3.3. Relationships between Properties of Thermally Modified Wood and Its Color

Properties of thermally modified wood correlated with color parameters were selected based on their identified marked dependence on modification temperature and high variability in the analyzed range of temperatures. These criteria are met by WL (Table 3), TS_L_ and MOE_R_ (Table 2). The dependence between WL and parameters ∆E and ∆L is well described by an exponential function (R^2^ = 0.92), while that between MOE_R_ and TS_L_ is well described by linear functions (R^2^ = 0.92 and R^2^ = 0.80–0.82). This is shown in Figure 5.

The existence of distinct functional dependencies between mass loss, longitudinal tensile strength and modulus of elasticity at radial compression of thermally modified wood and total color difference (∆E) and the difference in lightness (∆L) indicates that the above-mentioned parameters are good descriptors of analyzed properties; they may be considered indicators of changes taking place in wood during thermal modification. Based on color parameters, the technological quality of thermally modified wood may be assessed.

### 3.4. Chemical Analysis

Thermal modification caused mass loss in wood, which was connected with changes in the content of its components, as shown by the data presented in Table 3 and Figure 6.

The increase of temperatures used for wood modification resulted in higher values of mass loss in modified wood samples, from 0.14% (130 °C) to 7.00% (220 °C), as well as an increase in the total carbon content, from 47.43% (130 °C) to 50.33% (220 °C), which is consistent with literature data [42,43]. The content of extractives soluble in ethanol was similar for all variants of modification temperatures and comparable to the content in control wood. A slight decrease in extractives contents in the case of wood modified at 220 °C was observed in comparison to control wood and wood modified at lower temperatures. The results described by Esteves et al. [44,45] indicated that the content of extractives increased substantially upon the thermal modification of eucalypt and pine wood, but after wood reached a mass loss of 6%, the extractive contents exhibited a decreasing trend. In wood exposed to high temperatures, certain extractives were decomposed, but at the same time, new extractives were created, caused by the decomposition of main wood ingredients [46]. According to literature data, the increase, followed by a decrease, in extractive amounts in thermally modified wood suggests that there is an equilibrium between degradation/volatilization and that the appearance of new extractable compounds comes from polysaccharides and lignin degradation. As the temperature increases, the new compounds formed as a result of the degradation of carbohydrates and lignin degrades to volatiles, causing a decrease in extractives contents [45].

Table 3 presents the mean values and standard deviation (±σ_n−1_) of the recorded parameters.

**Table 3 materials-15-05776-t003:** Mass loss and total carbon content in modified wood.

	Temperatures of Modification (°C)				
	130	160	190	220	Control
Mass loss (%)	0.140 ± 0.002	0.690 ± 0.010	2.060 ± 0.110	7.000 ± 0.150	-
Total C content (%)	47.433 ± 0.311	47.742 ± 0.620	48.201 ± 0.161	50.327 ± 0.059	47.757 ± 0.069

The modification of pine wood caused changes in polysaccharide amounts, mainly by the degradation of hemicelluloses. Contents of cellulose and lignin in wood modified in the temperature range of 130–190 °C were similar and comparable to those in control wood. In turn, wood modified at 220 °C showed a relative increase in the content of cellulose (about 21%) and lignin (about 17%) compared to control wood. The increase in the temperature of wood modification caused a decrease of holocellulose content, which changed from 87.5% (control wood) to 63.6% (wood modified at 220 °C). However, significant changes were observed in hemicelluloses content. With increasing temperatures of the modification process, the content of hemicelluloses decreased, which was especially evident in the case of wood modified at 220 °C, where the hemicelluloses content was 6.3% compared to 42.3% determined for the control wood samples. The reduction of hemicelluloses content in wood modified at the highest temperatures was about 85% compared to control wood.

The obtained results indicated that hemicelluloses were the least stable ingredient of thermal modified wood, which is consistent with literature data [46,47,48,49]. The degradation of hemicelluloses was significantly faster compared to cellulose, which is connected with their amorphous nature and lower molecular weight [47,50].

The changes in wood structure caused by thermal modification were determined by ATR-FTIR analysis, the results of which in the form of spectra are presented in Figure 7.

The most important changes in the spectra of thermally modified wood compared to control wood were observed in the bands at 1735 cm^−1^ attributed to the C=O ester non-conjugated carbonyl group or carboxylic acid from the degradation of hemicelluloses and at 1660 cm^−1^ coming from C-O in quinones, coupled with C=O stretching in various groups, the intensities of transmittance of which decrease with an increasing temperature of modification [51,52]. These changes can be connected with the cleavage of acetyl groups, mainly in hemicelluloses. The intensities of the band at 1510 cm^−1^, ascribed to the C=C stretching of the aromatic skeletal vibrations in lignin in the spectra of modified wood, especially at higher temperatures, showed a reduction compared to control wood. The changes in the intensities of the band at 1510 cm^−1^ may be caused by the cleavage of methoxyl groups. The decrease of intensities in the band at 1030 cm^−1^ (C-O ester stretching vibrations in methoxyl and β-O-4 linkages in lignin) in the spectra of modified wood compared to control samples may be caused by the cleavage of β-O-4 linkage and methoxylates in lignin [46]. Moreover, other characteristic bands of lignin at 1460 cm^−1^ and 1270 cm^−1^ decreased in the spectra of modified wood, which confirmed that lignin degradation also occurred during thermal modification. In turn, the increase of intensities in the band at 1600 cm^−1^ in wood after heating was connected with an increase in the aromatic rings in wood components after their degradation [52]. In the spectra of modified wood, the reduction in the intensities of the broad -OH band at 3350 cm^−1^ was observed in comparison to control wood, which may be connected with the loss of -OH groups following wood dehydration upon heating [47]. The results of the ATR-FTIR analysis showed that all wood components were degraded during thermal modification.

## 4. Conclusions

The research of thermally modified Scots pine sapwood (*Pinus sylvestris* L.) showed that the total color difference ∆E and the difference in lightness ∆L and the chromatic coordinates ∆a are linear functions of modification temperature. A positive effect of thermal treatment was shown on the modulus of elasticity in the longitudinal, tangential and radial directions and a lack of such an effect on longitudinal compressive strength, as well as a negative effect of the thermal treatment on compressive strength perpendicular to grain, longitudinal tensile strength and impact strength. With an increase in modification temperature, a reduction is observed (to a greater or lesser extent) in the modulus of elasticity perpendicular to grain, compressive strength in the radial direction, longitudinal tensile strength and impact strength. Functional dependencies were found between mass loss, modulus of elasticity in the radial direction and longitudinal tensile strength, and total color difference and the difference in lightness. This makes it possible to determine these properties of thermally modified Scots pine sapwood following the ThermoWood^®^ procedure based on its color.

## Figures and Tables

**Figure 1 materials-15-05776-f001:**
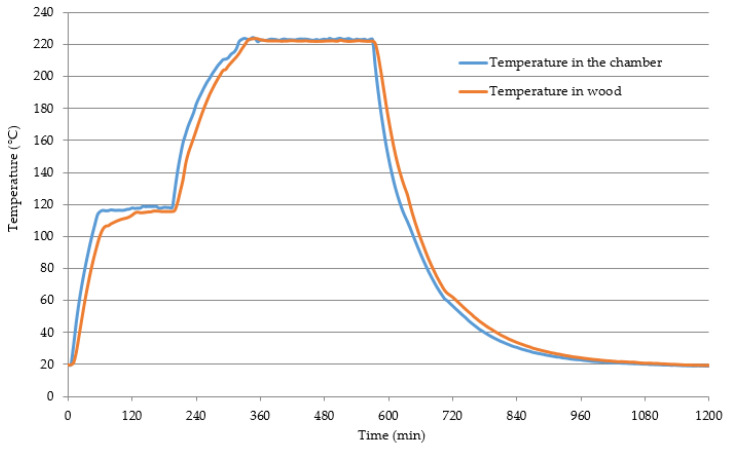
The course of the wood modification process (temperature 220 °C, time 4 h) [16,17].

**Figure 2 materials-15-05776-f002:**
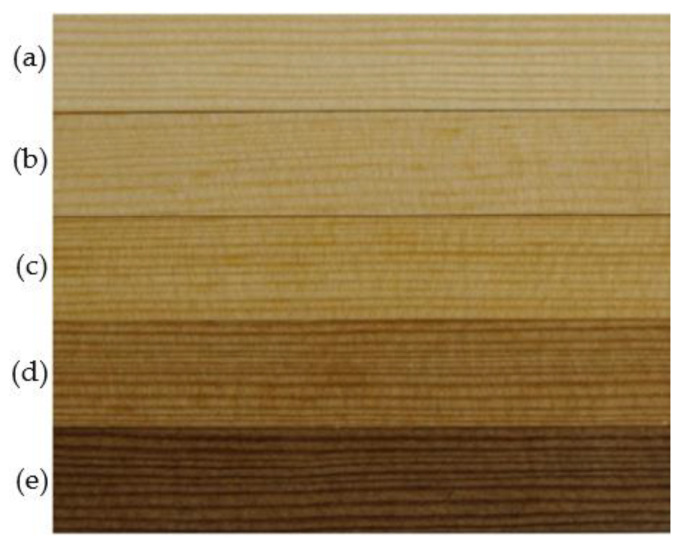
Sample surface—control (**a**) and modified wood (constant modification time—4 h) at a temperature of 130 °C (**b**); 160 °C (**c**); 190 °C (**d**); 220 °C (**e**).

**Figure 3 materials-15-05776-f003:**
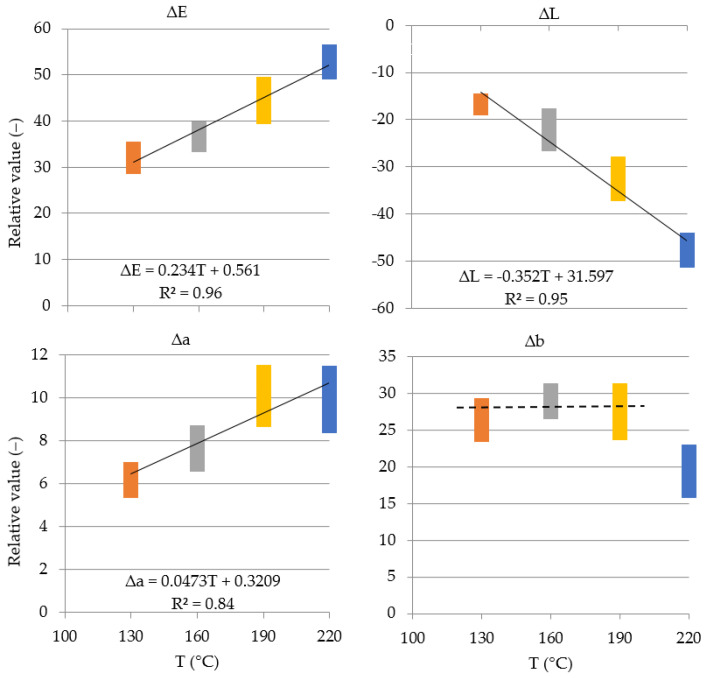
Changes in ∆E, ∆L, ∆a and ∆b in the function of modification temperature (T).

**Figure 4 materials-15-05776-f004:**
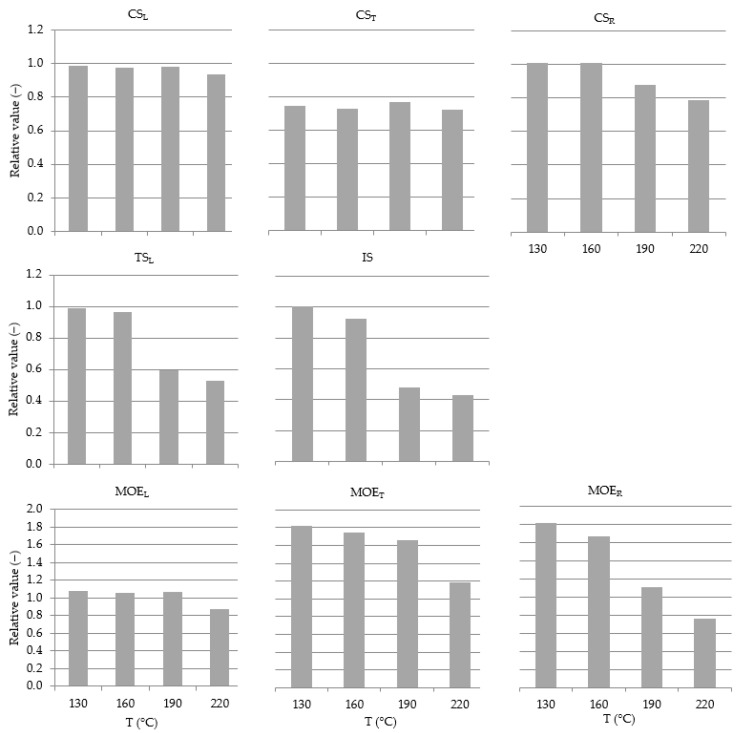
Relation-to-control value (in proportion) versus T (°C). Each column represents the mean value of *n* samples. IS, *n* = 20; CS_L_, *n* = 10; CS_T_, *n* = 10; CS_R_, *n* = 10; TS_L_, *n* = 10; MOE_L_, *n* = 10; MOE_T_, *n* = 10; MOE_R_, *n* = 10.

**Figure 5 materials-15-05776-f005:**
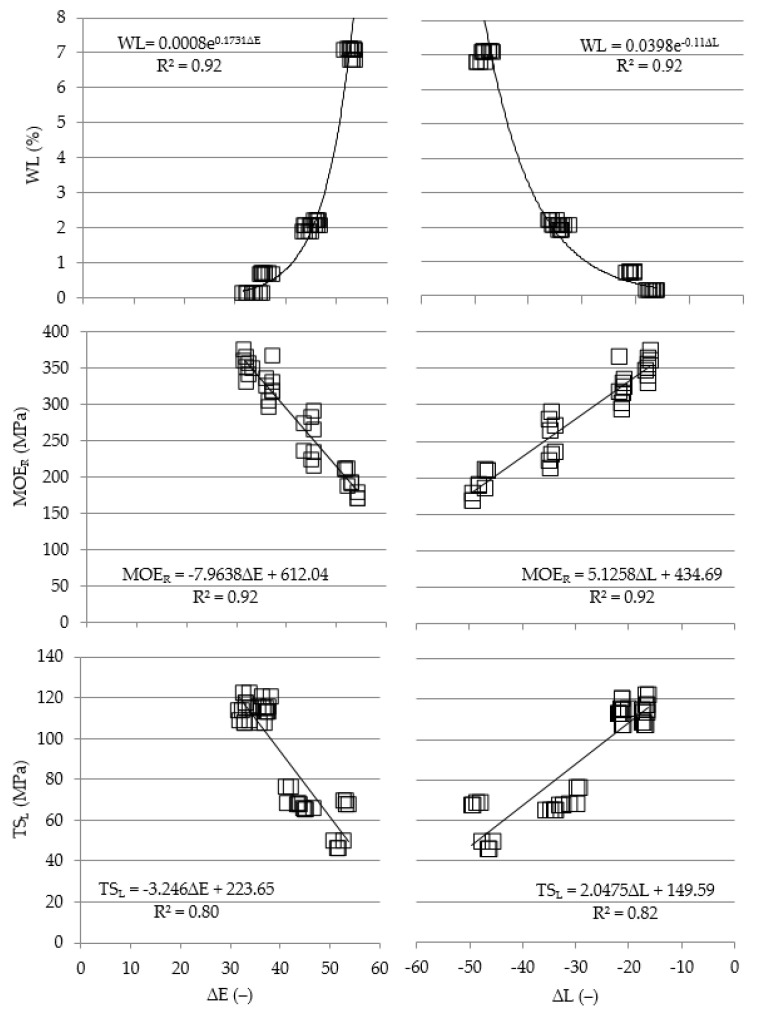
Values of WL, MOE_R_ and TS_L_ in the function of total color difference (∆E) and difference in lightness (∆L).

**Figure 6 materials-15-05776-f006:**
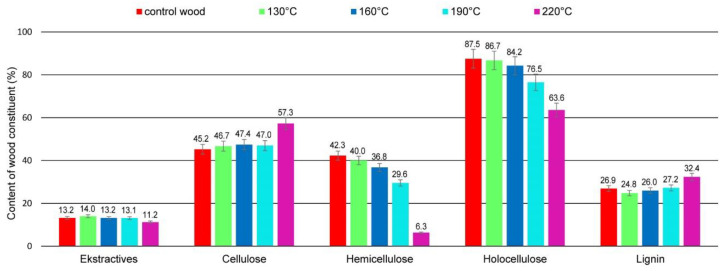
Content of extractives and main components in modified wood.

**Figure 7 materials-15-05776-f007:**
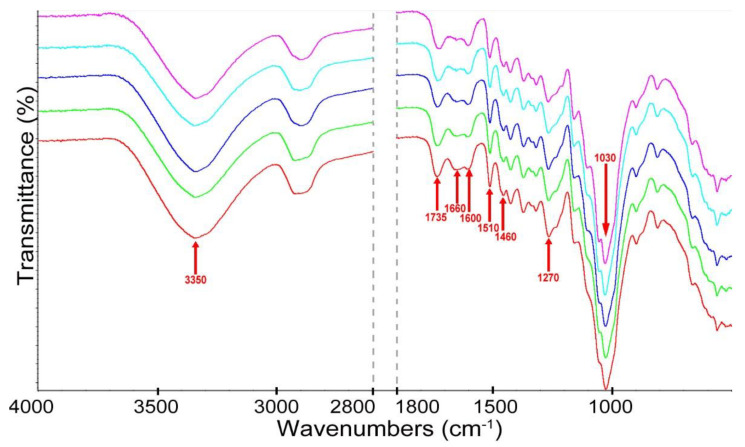
The ATR-FTIR spectra of control wood (–), wood modified at 130 °C (–), 160 °C (–), 190 °C (–) and 220 °C (–).

**Table 1 materials-15-05776-t001:** Parameters of color change in modified wood at different temperature and at constant time.

Temperature of Modification T (°C)	Parameters of Color (–)			
	∆L	∆a	∆b	∆E
130	−16.87 ± 0.49	6.32 ± 0.28	26.75 ± 0.96	32.25 ± 0.93
160	−21.71 ± 1.59	7.66 ± 0.38	28.50 ± 0.67	36.66 ± 0.97
190	−33.78 ± 1.51	10.22 ± 0.50	27.14 ± 1.30	44.57 ± 1.37
220	−47.93 ± 1.21	10.19 ± 0.49	20.18 ± 1.02	53.01 ± 1.00
Control *	−13.07 ± 0.71	4.90 ± 0.31	23.56 ± 0.71	27.38 ± 0.87

* Values of control wood parameters are mean values from the color measurements of all control (twin) samples.

**Table 2 materials-15-05776-t002:** Mechanical properties of wood.

Mechanical Properties	Temperatures of Modification (°C)				
	130	160	190	220	Control *
CS_L_ (MPa)	75.2 ± 4.0	75.6 ± 6.5	73.7 ± 4.2	67.9 ± 3.4	75.3 ± 3.8
CS_T_ (MPa)	5.2 ± 0.3	5.2 ± 0.2	5.4 ± 0.5	5.4 ± 0.5	7.1 ± 0.6
CS_R_ (MPa)	3.5 ± 0.2	3,5 ± 0.2	3.0 ± 0.4	2.7 ± 0.3	3.5 ± 0.2
TS_L_ (MPa)	114.1 ± 5.4	114.2 ± 4.1	68.7 ± 4.4	58.4 ± 11.9	115.0 ± 10.1
MOE_L_ (GPa)	19.3 ± 2.0	18.8 ± 0.7	18.5 ± 0.9	16.2 ± 0.7	17.9 ± 1.1
MOE_T_ (MPa)	750.0 ± 35.4	760.0 ± 25.8	720.0 ± 31.5	550.0 ± 41.5	440.0 ± 29.3
MOE_R_ (MPa)	350.0 ± 13.9	320.0 ± 21.7	250.0 ± 29.3	190.0 ± 17.1	220.0 ± 15.8
IS (J/cm^2^)	3.84 ± 0.43	4.27 ± 0.40	1,64 ± 0.38	1.54 ± 0.27	3.97 ± 0.61

* Mean values of a given property were determined for all control wood (control) samples.

## Data Availability

The data presented in this study are available on request from the corresponding author.

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
