# Peer review of "Color as an Indicator of Properties in Thermally Modified Scots Pine Sapwood"

_materials, 2022, doi:10.3390/ma15165776_

Round 1

Reviewer 1 Report

L50-60. Interesting speculation but do you have any specific examples of the ‘hinderance’?

L72-76 Any specific gaps in these two references that you seek to fill in your study?

L81-83 You are using only one modification types on only one species – will your results be more broadly applicable?

L87 with reference to previous question, is this only sapwood? If so, this seems very controlled but very narrow: clear sapwood of one board of one species, modified using one process…

L102 – do you have a reference for the thermowood process? Are the temperatures you chose as treatments similar to those used in commercial thermowood?

L115 were all the replicates done in one batch (for each treatment)? If, you are again minimizing variation but also, I suggest, scope of inference

L131 no measurement of MOR? Seems like a very important parameter – why not measured?

L137 – 146. I think I understand your logic but its also unrealistic, given that the control wood would never be the same moisture content in service. I think you should acknowledge the bias this introduces.

L178 verb tense is active, whereas it is passive voice elsewhere. Science often written in past tense, but at least be consistent.

L211 This is where it is hard for you to speculate, given the lack of variation that is included in your study but could occur in reality. How much color variation would you expect from batch to batch, based on slight variations in process or feedstock for example?

L271 Is this already a known effect – modification potentially greatly increasing MOE? If so, there should be reference to other publications with similar findings, or, if not, discussion of this is needed.

Also, given that MOE and MOR are correlated in normal wood (and this correlation is used to for prediction in commercial material), if is a shame that no MOR measurements were done to see how this correlation is affected

L242-297 Given that you have Table 2 and Figure 4, I think you can take out some of the text descriptions of the data and highlight only the noteworthy trends or effects.

L314 Again, hard for you to justify that inference bases on the very limited variation allowed in your experiment.  And is an 80% R-squared all that useful?

How is the chemical analysis (e.g. section 3.4) relevant to this paper? I can understand that certain chemical changes could result in, or be correlated to, certain color/mechanical properties but I don’t see that you make such connections.

L405 This ‘possible’ was know before, as you reference in the introduction and background. How has this paper changed the state of knowledge?

Author Response

Materials-1813667     

We would like to thank the Reviewer for your insightful and substantive comments. We very much appreciate your suggestions, questions and reservations regarding the work submitted for review, which have been very helpful in improving the manuscript. Below you will find our answers to your points.

The Authors fully agree with the suggested changes in the text and the Reviewer's comments.

Comments and Suggestions for Authors

L50-60. Interesting speculation but do you have any specific examples of the ‘hinderance’?

L72-76 Any specific gaps in these two references that you seek to fill in your study?

The higher the modification temperature, the darker the wood becomes. At the same time its equilibrium moisture content decreases. This is manifested in higher strength values recorded​​in the tests. If this is not taken into account, a controversial observation can be drawn that wood modified at low temperature (light-colored wood – light braun and with relatively high moisture content) has a lower strength than wood modified at high temperature (very dark wood – dark brown and with low moisture content), and the "strength – color" relationship will be increasing. In other words, wood subjected to intensive thermal treatment, the effect of which is the degradation of all chemical components and strong darkening, would be characterized by higher strength than poorly/slightly  modified or unmodified wood? In this sense, the study is part of the discussion whether color may be a predictor of strength of heat-treated wood and of properties of heat-treated versus unmodified wood.

L81-83 You are using only one modification types on only one species – will your results be more broadly applicable?

The scope of inference and conclusions have been limited.

L87 with reference to previous question, is this only sapwood? If so, this seems very controlled but very narrow: clear sapwood of one board of one species, modified using one process…

The title was changed to:

Color as an indicator of properties in thermally modified Scots pine sapwood”;

the scope of inference and conclusions have been limited.

L102 – do you have a reference for the thermowood process? Are the temperatures you chose as treatments similar to those used in commercial thermowood?

We have added literature reports.

With reference to the above comments, I would like to mention:

The aim of this study was to determine whether color may be a predictor of properties of thermally modified Scots pine sapwood (L81-83, L87). In order to produce wood samples of different colors 4 different temperatures were applied. The modification process was run following the commonly used  ThermoWood® procedure, extended to incorporate the range of temperatures to 130 and 160°C (L102).

This aim of the study determined the selection of the material and the adopted research methodology. In order to minimize the effect of factors related to the anatomical structure of wood at the level of macro-, micro- and ultrastructure on the resulting properties of thermally modified wood and unmodified wood it was decided to conduct experiments on possibly the most homogeneous and defect-free material (L87). For this reason, wood for analyses came from the same zone of the cross-section and height of the stem. Tested wood properties were determined on twin samples of identical moisture content.

L115 were all the replicates done in one batch (for each treatment)? If, you are again minimizing variation but also, I suggest, scope of inference

The scope of inference and conclusions have been limited.

L131 no measurement of MOR? Seems like a very important parameter – why not measured?

Out of many available static loading tests 2 tests were selected, in which the simplest stress conditions occur, i.e. linear tension or compression. This was done to simplify the analyses, in contrast to MOR measurements, where a complex state of stresses occurs, comprising at the same time compressive and tensile stresses.

L137 – 146. I think I understand your logic but its also unrealistic, given that the control wood would never be the same moisture content in service. I think you should acknowledge the bias this introduces.

Wood moisture content in the hygroscopic range to a considerable degree determines the mechanical properties of wood, generally wood strength decreases with an increase in moisture content. In order to obtain comparable values, the results obtained at different moisture contents need to be converted to the common comparable level. For this purpose, in the moisture content of 10-20% Bauschinger’s formula may be applied; however, it does not apply to thermally modified wood, for which equilibrium moisture content depending on treatment parameters may be around 3-5%. It is our opinion that in order to compare properties of wood (either of different origin or subjected to different types of treatment) strength needs to be determined on samples of identical moisture content. In turn, in recommendations concerning the application of thermally modified wood it needs to be remembered that it will have lower moisture content, which will influence the resulting preferences.

L178 verb tense is active, whereas it is passive voice elsewhere. Science often written in past tense, but at least be consistent.

We checked the entire manuscript and linguistic corrections were made.

L211 This is where it is hard for you to speculate, given the lack of variation that is included in your study but could occur in reality. How much color variation would you expect from batch to batch, based on slight variations in process or feedstock for example?

Tests were also conducted on sapwood of Scots pine of unknown origin (planks of 32 mm in thickness and 12 cm in width, sapwood density 510 kg/m3) modified under industrial practice conditions following the ThermoWood procedure at the temperature of 212°C. Color and tensile strength parallel to the grain were measured, providing the following results: ∆E from 48 to 56 (mean 52), TSL 36-56 MPa (mean 46 MPa).

L271 Is this already a known effect – modification potentially greatly increasing MOE? If so, there should be reference to other publications with similar findings, or, if not, discussion of this is needed.

An increase in MOE and at the same time rigidity and hardness of wood after heat treatment is a well-known fact. It is explained in literature by changes in the chemical structure of wood components, mainly an increase in the degree of crystallinity of cellulose. The manuscript was supplemented with literature references.

Also, given that MOE and MOR are correlated in normal wood (and this correlation is used to for prediction in commercial material), if is a shame that no MOR measurements were done to see how this correlation is affected

After compressive strength, bending strength (expressed by MOR) is the most frequently determined wood property. This test was not included in these experiments (as explained in the response to L131). We agree that knowledge on the relationships between MOE and MOR for modified wood would be interesting; however, it was not the aim of this study.

L242-297 Given that you have Table 2 and Figure 4, I think you can take out some of the text descriptions of the data and highlight only the noteworthy trends or effects.

It is possible, but it would only slightly reduce the length of the text. Moreover, Table 2 shows mean values of all control wood samples, while Fig. 4 shows values of modified wood in relation to respective values determined for twin samples of control wood, and not to mean values  from Table 2.

L314 Again, hard for you to justify that inference bases on the very limited variation allowed in your experiment.  And is an 80% R-squared all that useful?

The value of the coefficient of determination (R2=0.80) indicates good fitting of the linear function to the dependence between color and the temperature of wood modification.

How is the chemical analysis (e.g. section 3.4) relevant to this paper? I can understand that certain chemical changes could result in, or be correlated to, certain color/mechanical properties but I don’t see that you make such connections.

Results of presented experiments show, among other things, a considerable reduction of hemicelluloses content and an increase in the content of lignin in wood modified at 190 and 220°C. These changes in the chemical structure explain the reduction in TSL and IS, because mechanical properties of wood are positively correlated with the content of hemicelluloses (mainly pentosanes and hexosanes) and negatively correlated with lignin content. The respective explanation was included in the text.

High temperature treatment causes degradation of one or more chemical cell wall components (cellulose, lignin, hemicelluloses), which leads to changes in physico-mechanical properties of modified wood. These phenomena were confirmed by literature reports (References: 1-3; Corleto, R.; Gaff, M.; Niemz, P.; Kumar Sethy, A.;  Todaro, L.;  Ditommaso, G.;  Razaei, F.;  Sikora , A.;  Kaplan, L.;  Das, S.;  Kamboj, G.;  Gasparik, M.;  Kacik, F.;  Macku, J.,  Effect of thermal modification on properties and miling behaviour of African padauk (Pterocarpus soyauxii Taub.) wood.  J. Maters Res Technol. 2020; 9(4):9315-9327; Hirai, N.; Sobue, N.; Asano, I. Studies on piezoelectric effect of wood. IV. Effects of heat treatment on cellulose crystallities and piezoelectric effect of wood. Mokuzai Gakkaishi 1972, 18 (11), 535-542; Kymäläinen, M.; Mlouka, S.B.; Belt, T; Merk, V.; Liljeström, V.; Hänninen, T.; Uimonen, T.; Kostiainen, M.; Rautkari, L. Chemical, water vapour sorption and ultrastructural analysis of Scots pine wood thermally modified in high-pressure reactor under saturated steam. J Mater Sci 2018, 53, 3027–3037. DOI:10.1007/s10853-017-1714-1).

Also, changes in the color of wood subjected to high-temperature treatment are associated with changes in the degradation of the main components of the wood. Wood after thermal modification has a tendency to take on darker shades, which is associated with an increase in the phenolic group and stabilization of lignin during thermal modification (References: 41).

L405 This ‘possible’ was know before, as you reference in the introduction and background. How has this paper changed the state of knowledge?

Earlier studies determined relationships between wood properties and mass loss or with treatment temperature, as well as between wood color and temperature or sometimes the time of thermal treatment. Weaker or stronger linear or curvilinear correlations (0.41 <R2< 0.94) were found depending on the wood species and analyzed property. Reports that color measurements may be applied to assess quality of heat modified wood were based on the existence of a correlation between color parameters and intensity of thermal treatment, which is indicated by the loss of mass or the temperature of modification.

In the presented study it was important to determine functional dependencies making it possible to calculate values of certain mechanical properties and loss of  mass based on the total color difference ∆E and/or the difference in lightness ∆L.

We greatly appreciate the in-depth analysis and advice that have been a valuable contribution to our manuscript. We would like to thank you very much again for your review of our manuscript.

Reviewer 2 Report

The research showed that the thermally modified wood can be analyzed by the difference in colors in the wood samples in relation to their properties. The methodology used to determine the cores is correct by the CIELAB system. The work is relevant in the area and clearly showed that the observation of colors is directly related to the modification of wood properties. The work is well written and easy to read. The results are apparently correct, the researchers were able to demonstrate this in the work and recommend its publication.

Author Response

Materials-1813667

We would like to thank the Reviewer for yours positive opinion on our manuscript and the comment.

Comments and Suggestions for Authors

The research showed that the thermally modified wood can be analyzed by the difference in colors in the wood samples in relation to their properties. The methodology used to determine the cores is correct by the CIELAB system. The work is relevant in the area and clearly showed that the observation of colors is directly related to the modification of wood properties. The work is well written and easy to read. The results are apparently correct, the researchers were able to demonstrate this in the work and recommend its publication.

Reviewer 3 Report

accepted with minor revision

Author Response

Materials-1813667

We would like to thank you for your helpful and constructive comments and suggestions. We very much appreciate your suggestion, which has been very helpful in improving the manuscript. Below you will find our answer to your suggestion.

The Authors fully agree with the suggested changes in the text and the Reviewer's comments.

Comments and Suggestions for Authors

accepted with minor revision

The authors have determinated the dependencies between mechanical properties of modified wood and its color. Within its scope quantitative changes in color and chemical com position (mass loss, total carbon content, content of extractives and main components of wood) as well as mechanical properties (compressive strength along the grain, strength and modulus of elasticity in longitudinal tension tests, compression across the grain and impact resistance) of modified sapwood of Scots pine were determined. The paper is very interesting but it need minor revision to publish in this journal.

below are my comments:

  1. Authors are invited to ameliorated the title of the paper

The title was changed to:

Color as an indicator of properties in thermally modified Scots pine sapwood”.

  1. Materials 2.2.1. The course of the modification process is shown in Figure 1 on the example of a modification at a temperature of 220°C. why do you choice 220°C?

Figure 1 presents the course of the modification process at 220°C, since this is the temperature, at which the greatest changes were recorded in the observed parameters (color and properties of wood).

The boundary temperature for the wood modification process under commercial conditions (in industrial practice) is 220°C. At this temperature considerable changes occur in the structure, mechanical properties, as well as natural color of wood.

  1. Authors should ameliorated the 2.2.1 methods part

If remark 3 refers to remark 2, then in the opinion of the authors it is not necessary to present graphs of the course of the modification process for all temperatures, because the character of the course of temperature changes in time is analogous, the only difference is in the maximum temperature of treatment maintained for 4 hours.

  1. Methods and materials 2.2.3. part: The machine software apart from the graphic plotting of the test made it also possible to calculate the values of strength and modulus of elasticity (MOE).

Which softaware and how the values of strength and modulus of elasticity (MOE) were calculated?

The ZWICK Z050TH strength testing machine is equipped with the Test Expert program, which make possible a digital recording of the course of the test and its visualization in the form of a graph in the stress-strain system (Ϭ-ε). Strains were measured using a ZWICK 066550.02 extensometer.

Prior to the tensile test values of cross-section dimensions of samples and test parameters: loading speed, initial loading and the extensometer base were entered in the computer program.

Values of MOE were determined based on the graph of Ϭ-ε as a quotient of the increment in stresses to strains calculated based on the linear segment of the graph, i.e. from the formula E= Ϭ/ε.

Values of strength at compression and tension parallel to the grain were calculated as a quotient of force at failure to the surface area of the sample cross-section, while in the case of directions perpendicular to the grain it was as a quotient of force at the limit of proportionality to the surface area of the sample cross-section.

  1. The language needs to be improved in the text

We checked the entire manuscript and linguistic corrections were made.

  1. Results and discussion. Wood color, ligne 178-181, line 178-181, they lack references to justify various assertions.

The test in lines 178-181 refers to  Fig. 2, while the numerical data of measured color parameters are given in Table 1. Differences in values of ∆E definitely exceed 3 units are visible with the naked eye. The difference in color of 2-3 units is considered at the threshold for the perception of a color change (e.g. References: 30).

  1. Line 179- 180, The most marked change is observed for samples exposed to the temperature of 220° C. why?justify

The wood color after heat treatment depending on the temperature varies from light to dark brown, caused by the formation of quinones or the carmellization of holocellulose components. The higher the modification temperature, the more intensive the wood color darkening (References: 7,9). Thermal modification resulted in darkening of the color of wood, which is connected with degradation of wood components during high heat treatment. Literature data reported that changes of wood color showed high correlation with the levels of degradation of hemicelluloses and lignin. Also, temperatures and time of thermal modification have an effect on the wood color (References: 31; Sivonen, H.; Maunu, S.; Sundholm, F.; Jämsa, S.; Viitaniemi, P. Magnetic resonance studies of thermally modified wood. Holzforschung 2002, 56, 648–654;  Esteves, B.; Marques, A.; Domingos, I.; Pereira, H. Heat-induced colour changes of pine (Pinus pinaster) and eucalyptus (Eucalyptus globulus) wood. Wood Sci. Technol. 2008, 42, 369–384; Schneider, A. Investigations on the convection drying of lumber at extremely high temperatures. Part II: Drying degrade, changes in sorption, colour and strength of pine sapwood and beech wood at drying temperatures from 110 to 180°C. Holz Roh- Werkst. 1973, 31, 1998–2206). Therefore, wood modified at 220°C showed the darkest color, which is connected with an increase of lignin content in wood samples.

  1. Table 1. why did the values of the parameters Δa and ΔE are high at 190° C? while that of Δb is high at 160° C?

The maximum difference in the value of ∆b within the temperature range of 130-190°C is 1.75 units and it is not significant in terms of color change, because variation in color amounting to 2-3 units is considered to be the threshold of perception of color change. At the temperature of 220°C the value of ∆b decreases by approx. 7 units (wood color becomes less yellow).

∆E is the function of color parameters ∆a, ∆b, ∆L. Values of ∆E, ∆L, ∆a increase with an increase in the modification temperature. ∆E and ∆L reach maximum values at 220°C, while the value of ∆a at 220 and 190°C is the  same (maximum).

  1. Table 2, why CST values do not vary much?

The layers of late wood in annual rings are mainly responsible for compressive strength of wood in the tangential direction. Late wood exhibits greater mechanical strength, it is characterized an even 3-fold greater density and thicker cell walls compared to early wood. These wood traits are responsible for small changes in CST values.

  1. Table 2, are there link between the values of MOEL, MOET, MOER? Do these variations respect a law?

Generally MOEL is many times greater (even several dozen times) than the modulus of elasticity perpendicular the grain (e.g. Niemz, P.: Physik des Holzes und der Holzwerkstoffe. DRW-Verlag Weinbrenner GmbH&Co.1993). In the case of softwood species MOET assumes higher values than MOER, while in the case of hardwood species this dependence may be opposite. Experimentally determined values of MOEL depend significantly on the deflection angle between the direction of the acting force (tensile force, compressive force) and the direction of the grain
(e.g. Langendorf, G. et al.: Rohholz. Hanser Fachbuchverlag, Leipzig, 1990), i.e. on the appropriate execution of the test, mainly on the correct anatomical orientation of the sample, while at the level of wood ultrastructure – on the microfibrilar angle (MFA) in the S2 layer of the secondary cell wall.

  1. Specify axes on the diagrams in Figure 4

The labelling of the axis ”Relative values” was supplemented. Figure 4 presents relative values of tested properties (values determined for modified wood in relation to values for unmodified wood) versus temperature. For this reason the ordinate axis (Y) is not ascribed any units.

  1. Line 283-285: No significant change was observed in longitudinal tensile strength (TSL) of wood 283 modified at a temperature of 130 and 160° C compared to unmodified wood, while modification at 190 and 220° C caused a considerable reduction of TSL by 40 and 50%, respectively (Figure 4). Why?

Mechanical properties of wood depend on its chemical structure and thermal processing causes changes in wood cell wall chemistry. Reduction of TSL values in the case of thermal modification of pine wood at a temperature of 190 and 220°C is consistent with the reports of other researchers, who stated that treatment at a temperature above 150°C causes a reduction of tensile strength parallel to the grain and bending strength of wood (References: 11,12).

Results of our studies show considerable reductions in the content of hemicelluloses and an increase in the share of lignin in wood thermally modified at 190 and 220°C. These changes in the chemical structure may be considered the cause for the decrease in TSL, since mechanical properties of wood are positively correlated with the contents of pentosanes and hexosanes and negatively correlated with lignin (References: 10).

  1. what did x and y of equations in figures 5 represent? . is it normal to keep x and y?

The formula for the function on the graph, i.e. the use of x and y results from the computer program (y – dependent variable of the function: WL, MOEL, TSR; x – independent variable of the function: ∆E, ∆L).

The formula of the function in graphs 3 and 5 was changed.

  1. Authors should insert TG and DTG curves to accompany their discussion of part 3.4 and compare these values (of TG and DTG) with the diagrams in Figure 6

  1. The results of the ATR-FTIR analysis showed that all wood components were degraded during thermal modification. Accompain it with TG and DTG curves

Thank you for your suggestion. Unfortunately, we did not perform the TG analysis, because the quantitative changes of the main components of the tested wood were determined by the chemical composition analysis (determination of the content of cellulose, lignin, hemicelluloses) and FTIR analysis. However, if the reviewer finds that the TG analysis is necessary, we will try to carry out such an analysis and attach its results to the manuscript.

  1. specify the axes of figure 7

The y axis is transmittance (%) and the x axis is the wavenumber (cm-1).

  1. authors should synthetized the conclusion

We changed the Conclusions part in the manuscript.

We greatly appreciate the in-depth analysis and advice that have been a valuable contribution to our manuscript. We would like to thank you very much again for your review of our manuscript.

Round 2

Reviewer 1 Report

You responded to my comments in the 'authors' response' file but I don't see any changes related to my suggestions in the revised manuscript. To repeat, my major concerns are the lack of justification for this research (others have looked at using color to estimate modification) and the limited scope of inference. 

Also, it is incorrect to say that properties (in general) are negatively correlated to lignin content. Some properties maybe...

Author Response

We would like to thank the Reviewer for your reassess. Below you will find our answers to your comments.

Comments and Suggestions for Authors

You responded to my comments in the 'authors' response' file but I don't see any changes related to my suggestions in the revised manuscript. To repeat, my major concerns are the lack of justification for this research (others have looked at using color to estimate modification) and the limited scope of inference.

Also, it is incorrect to say that properties (in general) are negatively correlated to lignin content. Some properties maybe...

In the (previously) revised manuscript, the following corrections were made in accordance with the Reviewer's comments:

The scope of inference and conclusions have been limited do the Scots pine sapwood.

The title was changed.

We have added literature reports.

We have added reference for the ThermoWood® procedure.

The manuscript was supplemented with literature references with regard an increase in MOE of wood after heat treatment.

In a revised (current) version of manuscript:

The Materials part expanded to justify the selection of research material.

Additionally, the conclusions was narrowed down to the sapwood of modified pine according to the ThermoWood® procedure.

The introduction has been supplemented/changed in the context of explaining the rationale for undertaking this research. The purpose, scope of the work and conclusions were limited to modified wood according to the ThermoWood® procedure and Scots pine sapwood, which is also indicated in Materials and Methods.

In the context of previous research (by other authors) describing the relationships between the mechanical properties of thermally modified wood and its color, the novelty of the presented study is an indication of the existence of such relationships also between WL, MOER, TSL, and total color difference and the difference in lightness.

The authors agree with the Reviewer's comment. In response and in the text on the line 310 was missing "heat modified wood", which obviously changed the meaning of the sentence. We are sorry for this neglect. Corrected and supplemented text in lines 330.
